# Pharmacogenetics Approach for the Improvement of COVID-19 Treatment

**DOI:** 10.3390/v13030413

**Published:** 2021-03-05

**Authors:** Ingrid Fricke-Galindo, Ramcés Falfán-Valencia

**Affiliations:** HLA Laboratory, Instituto Nacional de Enfermedades Respiratorias Ismael Cosío Villegas, Mexico City 14080, Mexico; ingrid_fg@yahoo.com.mx

**Keywords:** COVID-19, pharmacogenetics, CYP3A4, CYP2D6, ABCB1, NR1I2

## Abstract

The treatment of coronavirus disease 2019 (COVID-19) has been a challenge. The efficacy of several drugs has been evaluated and variability in drug response has been observed. Pharmacogenetics could explain this variation and improve patients’ outcomes with this complex disease; nevertheless, several disease-related issues must be carefully reviewed in the pharmacogenetic study of COVID-19 treatment. We aimed to describe the pharmacogenetic variants reported for drugs used for COVID-19 treatment (remdesivir, oseltamivir, lopinavir, ritonavir, azithromycin, chloroquine, hydroxychloroquine, ivermectin, and dexamethasone). In addition, other factors relevant to the design of pharmacogenetic studies were mentioned. Variants in *CYP3A4*, *CYP3A5*, *CYP2C8*, *CY2D6*, *ABCB1*, *ABCC2*, and *SLCO1B1*, among other variants, could be included in pharmacogenetic studies of COVID-19 treatment. Besides, nongenetic factors such as drug–drug interactions and inflammation should be considered in the search for personalized therapy of COVID-19.

## 1. Introduction

Coronavirus disease 2019 (COVID-19) is a complex disorder affecting several organ systems caused by the severe acute respiratory syndrome coronavirus 2 (SARS-CoV-2). In most cases, this disease clinically manifests with self-limiting mild-to-moderate symptoms of an upper respiratory tract infection, as well as general symptoms such as myalgia and fatigue. In severely affected patients, an uncontrolled immune response occurs, leading to an increase of pro-inflammatory cytokines and chemokines, and hospital and ICU care are required [1]. In severe COVID-19 patients, complications such as acute kidney injury, renal failure, myocardial injury, liver dysfunction, blood leukocyte abnormalities, septic shock, and disseminated intravascular coagulation have been described [2].

The mortality rate of COVID-19 varies among countries and the clinical conditions of the patients. The higher death rates have been associated with age, male gender, ICU care requirements, obesity, and chronic diseases (mainly oncologic, cardiovascular, metabolic, and neurodegenerative diseases) [3].

To date, there is a lack of a completely effective drug to treat COVID-19. The available treatment of COVID-19 is mainly symptomatic and is based on disease severity; however, several antimicrobials have been used for the disease treatment. The use of antiviral agents (remdesivir, lopinavir/ritonavir, oseltamivir), antibiotics (azithromycin), antiparasitics (chloroquine, hydroxychloroquine, ivermectin), and corticosteroids (dexamethasone) have been reported in the literature [4].

The evidence of the COVID-19 treatments’ effectiveness and recommendations for prescribing remains controversial, and information from clinical trials is continuously generated. Nevertheless, a lack of drug response or the occurrence of adverse drug reactions (ADR) in specific patients has been observed. For instance, some patients with COVID-19 treated with lopinavir/ritonavir have presented diarrhea, while others reported nausea and vomiting [5].

In this sense, pharmacogenetics could explain the inter-individual variability on drug response based on the genetic of COVID-19 patients [6]. Variants in genes encoding drug-metabolizing enzymes, transporters, or receptors have been reported, and they could provide the insight to achieve a personalized therapy leading to a better outcome of this emerged disease (Figure 1) [7,8,9]; nevertheless, several disease-related issues must be carefully reviewed in the pharmacogenetic study of COVID-19 treatment. Therefore, we aimed to describe the pharmacogenetic variants reported for drugs used for COVID-19 treatment. Besides, other factors relevant to the design of pharmacogenetic studies in this regard were mentioned.

We performed a search in pharmacogenomic databases (e.g., Pubmed, PharmGKB, Pharmacogene Variation Consortium, CYPalleles, UGTalleles), and pharmacogenomic guidelines (Clinical Pharmacogenetics Implementation Consortium, U.S. Food and Drug Administration, European Medicines Agency) about available information of genetic variants related to the drugs currently prescribed for the COVID-19 treatment. Information about pharmacogenes’ expression in inflammatory diseases and pharmacologic interactions, relevant to the enzymes and transporters identified for COVID-19 treatment, was investigated.

## 2. Current Drugs Employed in the COVID-19 Treatment

### 2.1. Remdesivir

Remdesivir (GS-441524) is a monophosphoramidate nucleoside analog pro-drug developed initially to treat Ebola virus disease. It binds to the viral RNA-dependent RNA polymerase, inhibiting viral replication through premature RNA transcription termination [10]. Studies in animal models have reported some efficacy for this drug in treating coronavirus diseases such as SARS-CoV-2 and the Middle East Respiratory Syndrome coronavirus (MERS-CoV) [11,12]. Data from clinical trials have shown low-certainty evidence that remdesivir may be useful in reducing time to clinical improvement, reducing severe adverse events, and decreasing mortality in patients with severe COVID-19 [13,14].

Remdesivir is converted into its active triphosphate form, GS-443902, through metabolic conversion in cells and tissues. CYP2C8, CYP2D6, and CYP3A4 are involved in the metabolism of remdesivir, but it is considered that it is predominantly metabolized by hydrolases [15,16]. Several variants are described for the *CYP* genes with a relevant impact on the enzyme activity (Table 1). 

*CYP2C8* presents a low genetic variation, but alleles with no function have been well described [17]. Contrary, according to the Pharmacogene Variation Consortium (pharmvar.org, accessed on 19 February 2021), more than 141 *CYP2D6* alleles are identified with different enzyme activity impact. For instance, there are variants related to a lack of protein activity (e.g., *CYP2D6*3*, **4*, **6*); while *CYP2D6*10*, **17*, **29* or **41* are associated with reduced enzyme activity, and the duplication or multiplication of active alleles (e.g., *CYP2D6*1xN*, **2xN*) are related to increased activity of the enzyme [18]. The frequencies of *CYP2D6* variants present a significant interethnic variability, and some variants with an impaired activity can be commonly found in some populations, which can lead to differences in response to CYP2D6 substrates. For example, the *CYP2D6*4* allele frequency is higher among Caucasians than in other ethnic groups, while *CYP2D6*10* is shared among East Asians, *CYP2D6*41* and duplication/multiplication of active alleles in Middle Easterns, and *CYP2D6*17* in Black Africans [19]. The combination of *CYP2D6* alleles could predict the metabolic phenotype of a subject. For instance, individuals carrying two null alleles can be classified as poor metabolizers; those with one functional allele and one null allele as intermediate metabolizers; while the presence of two functional alleles classify the individuals as extensive metabolizers, and the phenotype ultrarapid metabolizers can be assigned if the functional alleles are duplicated or multiplied [20].

CYP3A4 is abundantly expressed in the liver of the majority of individuals. To date, more than 34 allelic variants have been reported, but the clinical impact of some of them remains unknown [17]. However, of considerable relevance in the COVID-19 treatment is that CYP3A4 presents a cytokine-mediated down-regulation in the course of the inflammatory response via JAK/STAT pathway, mainly by the interleukin-6 (IL-6) [21].

In addition, remdesivir has been observed to be a substrate of the organic anion transporting polypeptide 1B1 (OATP1B1) and P-glycoprotein (P-gp) [16,22]. The OATP1B1 transporter is a member of the transmembrane family transport proteins responsible for the uptake of substances into the cells of various organs, mainly of the liver [23]. It is encoded by the *SLCO1B1* gene in which several variants with impact in drug disposition have been identified. For instance, the rs2306283 c.388A > G is associated with a decreased transporter function and is commonly found in Africans, Asians, and Caucasians. Other variants present a low frequency among different populations but have been related to a decreased function of the transporters *SLCO1B1* rs56101265, rs56061388, rs72559745, rs4149056, rs72559746, rs55901008, rs59502379, and rs56199088 [24].

P-gp is an efflux pump encoded by the *ABCB1* gene, a member of the superfamily of the adenosine triphosphate (ATP)-binding cassette (ABC) genes. This transporter is expressed in the liver, small intestine and colon, kidney, and blood-brain barrier. P-gp putatively plays a role in viral resistance and trafficking cytokines and enveloped viruses [25]. Several *ABCB1* variants have been reported, however the rs1128503 c.1236C > T, rs2032582 c.2677G > T/A, and rs1045642 c.3435C > T have been of great relevance in pharmacogenetics studies. These three variants are located in the gene coding region and are in linkage disequilibrium; the main effects of the nucleotide changes in the protein have been related to a variation on transporter expression levels and an altered activity due to the c.3435C > T and c.2677G > T/A variants, respectively [26,27]. As will be described below, several drugs are P-gp substrates, and some of them can act as an inhibitor of the transporter modifying the disposition of the other drugs.

### 2.2. Lopinavir/Ritonavir

Lopinavir is a human immunodeficiency virus (HIV) protease inhibitor, which can suppress viral replication in combination with ritonavir. Lopinavir is rapidly and extensively metabolized by the CYP3A4 enzyme [28]. It is co-administrated with a low dose of ritonavir, a potent inhibitor of CYP3A4, to increase the plasma concentrations of lopinavir to achieve antiretroviral activity [29,30]. Besides the inhibition of CYP3A4, ritonavir also inhibits the P-gp in the gut wall, improving lopinavir absorption [31]. Therefore, the *CYP3A4* and *ABCB1* variants above described could also impact the pharmacokinetic and pharmacodynamics for lopinavir and ritonavir.

In addition to *CYP3A4* and *ABCB1*, the impact of variants in *CYP3A5*, *ABCC2*, and *SLCO1B1* in plasma concentrations of lopinavir have also been evaluated in different pharmacogenetic studies, but controversial results were observed. For instance, the *SLCO1B1* c.521T > C rs4149056 variant has been related to a reduced transport activity in vivo and variations of lopinavir plasma concentrations [32] and clearance [33]. The C allele’s homozygous state was associated with 37% lower clearance and 14% for the heterozygous condition [33]. Additionally, a reduced dosage requirement of lopinavir/ritonavir in patients with *CYP3A4*22/*22*, alone or in combination with *SLCO1B1* c.521CC, has been observed [34]. However, other studies have failed to found an association of *CYP3A4*, *CYP3A5*, *SLCO1B1*, and *ABCC2* variants with lopinavir/ritonavir plasma concentrations [35] and virologic outcome [36]. Likewise, genotypes and haplotypes of *ABCB1* variants could not predict lopinavir’s plasma concentrations in one study [37].

*ABCC2* (adenosine triphosphate (ATP)-binding cassette subfamily C member 2) encodes the human canalicular multispecific organic anion transporter, also called the multidrug resistance-associated protein 2 (MRP2). MRP2 is a specific nonbile acid organic anion transporter, which mediates the primary active export of conjugates of lipophilic compounds from cells using ATP [38]. Variants in *ABCC2* have been related to an altered transport of MRP2 substrates and the response to antiepileptic drugs [39]. The c.4544G > A rs8187710 variant in *ABCC2* has been associated with a higher accumulation of lopinavir in peripheral blood mononuclear cells of HIV-treated patients [40]. In contrast, a lower estimated glomerular filtration rate has been observed in patients carrying the T allele of *ABCC2* -24 C > T rs717620 compared to CC homozygotes [41]. 

CYP3A5, in conjunction with CYP3A4, accounts for approximately 30% of hepatic cytochrome P450, and nearly half of the commercialized drugs are metabolized by CYP3A enzymes. It is expressed in the liver and intestine, but this enzyme is the predominant form expressed in extrahepatic tissues. CYP3A5 is polymorphic, more than 25 variants in this gene have been reported, and some have been related to an altered enzymatic function (Table 1). *CYP3A5*3* is the most frequent among all populations, and it is a well-studied variant allele of *CYP3A5* in which an altered mRNA splicing occurs. Carriers of *CYP3A5*3/*3* genotypes are considered CYP3A5 non-expressors [42], and they could need lower dosages of CYP3A5 substrates such as lopinavir and ritonavir.

### 2.3. Oseltamivir

Oseltamivir is an ethyl ester pro-drug used for the treatment of influenza A and influenza B infection. To exert the antiviral effect, oseltamivir must be taken up by peptide transporter 1 (PepT1) and has to be converted to the active metabolite oseltamivir carboxylate through the enzyme carboxylesterase 1 (CES1) in the liver [43]. Although this antiviral has been used to treat COVID-19 [44,45], its effectiveness in this disease has not been wholly demonstrated [46].

Oseltamivir carboxylate inhibits viral neuraminidase (NEU2), blocking the release of progeny virions from infected cells and viral entry into uninfected cells [47]. Besides, oseltamivir is also a substrate of P-gp, which can eliminate the drug before being activated [48]. Wide inter-individual variability in the pharmacokinetics, response, and ADRs occurrence to oseltamivir has been observed [49]; thus, pharmacogenetic studies have investigated the *CES1* genetic variants’ relation with the variation observed in the oseltamivir treatment [43].

Variants in *CES1* have been found associated with variations in the pharmacokinetics of oseltamivir. The rs71647871 p.Gly143Glu has been related to variation in plasma concentration-time curve of oseltamivir [50], while a decrement in the antiviral drug bioactivation was found associated with the rs200707504 c.662A > G in CES1 [51].

Another study evaluated the association of oseltamivir ADRs with variants in *ABCB1*, *CES1*, *NEU2*, and *SLC15A1*, the gene encoding the transporter PepT1. Authors found that *ABCB1* rs1045642 was related to ADRs under the recessive model; the C allele was more frequently found among patients who did not present ADRs, while the T allele was predominant in the group of individuals who did report ADRs [52].

## 3. Pharmacogenetics of Azithromycin

Azithromycin is an azalide antimicrobial agent and structurally related to the macrolide erythromycin. It interferes with bacterial protein synthesis by binding to the 50S component of the 70S ribosomal subunit [53]. Due to its structural properties, azithromycin does not interact with cytochrome P450 enzymes, but it is a substrate of the transporters P-gp and MRP2 [54]. The interaction of azithromycin with P-gp suggests being the reason for its efficacy in the COVID-19 treatment and its synergistic effect when combined with hydroxychloroquine [55,56].

A pharmacogenetic study in 20 Chinese Han healthy volunteers found significant differences in Cmax and Tmax of azithromycin related to the *ABCB1* c.2677G > T/c.3435C > T genotype [57]. Likewise, a significant variation in Cmax and AUC related to *ABCB1* genotypes in Pakistani subjects has been observed [58].

## 4. Pharmacogenetics of Antiparasitics Used for COVID-19 Treatment

### 4.1. Chloroquine and Hydroxychloroquine

Chloroquine and hydroxychloroquine drugs are used in the treatment and prophylaxis of malaria. Both drugs are widely metabolized in the liver by CYP2C8, CYP3A4, CYP3A5, and, to a lesser extent, by CYP2D6 [59]. Genetic variants of these enzymes have been previously described (Table 1), and a study have suggested that *CYP2C8*, *CYP2C9,* and *CYP3A5* genetic variants influence chloroquine malaria treatment; however, there is not enough evidence of their impact on the chloroquine pharmacokinetics or response [8,60]. Transporters’ variants could be critical in the pharmacogenetics of chloroquine [61]. In this sense, a pharmacogenetic study, including Brazilian patients with malaria, reported an association of *SLCO2B1* and *SLCO1A2* variants with chloroquine response [62].

Regarding the pharmacogenetics of hydroxychloroquine, the *CYP2D6*10* variant has been found associated with the drug’s metabolic ratio [63], while variants in the gene of the transporter ABCA4 have been related with hydroxychloroquine toxicity [64].

In addition, international health agencies have warned about the use of chloroquine and hydroxychloroquine in patients with known glucose-6 phosphate dehydrogenase deficiency because hemolysis, and hemolytic anemia, can occur [65]. Although a recent study does not support this warning among African Americans [66], a case of hydroxychloroquine-induced hemolytic anemia in a 32-year-old patient of sub-Saharan African origin with glucose-6 phosphate dehydrogenase deficiency and COVID-19 has already been reported [67].

### 4.2. Ivermectin

Ivermectin is a semisynthetic derivative of avermectin B_1_ used as an antiparasitic drug. It is extensively metabolized by cytochrome P450 enzymes, predominantly by the CYP3A4 isoform, which converts ivermectin to at least 10 metabolites, most of them hydroxylated and demethylated derivatives [68,69].

Ivermectin also interacts with P-gp [70], and nonsense mutations in *ABCB1* has been related to severe neurologic ADRs induced by the antiparasitic [71], probably due to an altered function of the transporter at the blood–brain barrier that leads to toxic levels of ivermectin in the brain [72].

Besides, ivermectin’s interaction with OATP1A2 and OATP2B1 has been reported [73]; although, to the best of our knowledge, no pharmacogenetic studies evaluating the impact of *SLCO1A2* and *SLCO2B1* variants in the ivermectin therapy have been performed.

## 5. Pharmacogenetics of Corticosteroids Used for COVID-19 Treatment

### Dexamethasone

Dexamethasone is a glucocorticosteroid used to suppress cytokine release and inhibit lung infiltration by neutrophils and other leukocytes. CYP3A4 extensively metabolizes it into 6-hydroxydexamethasone and other metabolites in the human liver [74]. It is a substrate of P-gp, which is considered that contributes to steroid resistance [75]. In PharmGKB (pharmgkb.org, accessed on 19 February 2021), several variants are reported to influence the response and/or the toxicity to dexamethasone, including variants in *ABCB1* and other genes (Table 2). Nevertheless, the reported variants’ association should be taken with caution because patients treated with different drugs were included in the studies.

In addition, a study based on an expression quantitative trait loci (eQTL) analysis using >300 expression microarrays from lymphoblastoid cell lines in dexamethasone-treated and untreated cells derived from asthmatic subjects has identified significant pharmacogenetic loci including rs6504666 and rs1380657 (*SPATA20*), rs12891009 (*ACOT4*), rs2037925 and rs2836987 (*BRWD1*), rs1144764 (*ALG8*), and rs3793371 (*NAPRT1*) [76].

## 6. Considerations in Pharmacogenetic Studies of COVID-19 Treatment

COVID-19 has been a challenge for worldwide science and public health. Treatment recommendations with available drugs for this emerging disease have been established and adjusted since the beginning of the pandemic. Proposed drugs have been used for other infectious and non-infectious diseases, and therefore there is a background for the design of pharmacogenetic studies related to COVID-19 treatment [7]. However, other genetic and nongenetic factors should be taken into account, particularly for the treatment of COVID-19.

### 6.1. Drug-Drug Interactions

Most of the drugs used for the COVID-19 treatment are metabolized by CYP3A4 and are the substrate of P-gp and OATPB1 (Figure 2). Nevertheless, there are well-known drug-drug interactions related to the enzyme and the transporters, which should be considered in the study of drug response variability. For instance, relevant transporter interactions of chloroquine, hydroxychloroquine, ivermectin, ritonavir, lopinavir, favipiravir, and remdesivir with the ABCB1/P-gp, ABCG2/BCRP, and ABCC1/MRP1 exporters, as well as the OATP2B1 and OATP1A2 uptake transporters, have been reported [73].

As a complex disease, polypharmacy could be shared among patients with COVID-19, besides their routine treatment if they present chronic co-morbidities. In this sense, patients with COVID-19 could be treated with antimicrobial, anti-inflammatory, as well as chronic treatments (e.g., antidiabetic and antihypertensive agents), and the potential effect of the co-treatment in the drug response variability should be evaluated. In Table 3, several drugs considered as substrates, inhibitors, and inducers of CYP3A4, ABCB1, and OATPB1 are included. As observed, commonly prescribed drugs can be involved in drug interactions, including those for the COVID-19 treatment.

### 6.2. Orphane Nuclear Receptors

Regarding the inhibition or induction of enzymes and drug transporters, it is also worth mentioning that two genes encode relevant nuclear receptors that contribute to both auto-induction of drug clearance and drug–drug interactions in combined therapies. The orphan nuclear receptors PXR (pregnane X receptor, encoded by *NR1I2*) and CAR (constitutive androstane receptor, encoded by *NR1I3*) are xenobiotics’ sensors that mediate drug-induced changes by increasing transcription of genes involved in drug clearance and disposition. Therefore, genetic variability in these nuclear receptors could also contribute to the drugs’ response [77,78].

PXR is an approximately 434-amino acid, 50-kDa protein, mainly expressed in the liver and intestine. It contains an N-terminus region; a DNA binding domain consisting of two zinc fingers (amino acids 41–107); a hinge region (amino acids 107–141) and a ligand-binding domain containing the ligand-binding pocket, and a ligand-depending activation factor domain (amino acids 141–434) [79,80]. When a ligand binds to PXR, the receptor is activated, and it forms a heterodimer with 9-cis retinoic acid receptor RXR-alpha, which binds to the specific DNA region of the target genes to induce their expression [81]. PXR ligands include drugs, carcinogens, food additives, pesticides, and environmental pollutants. A wide variety of drugs that bind to PXR have been described, including antibiotics, anticancer drugs, antihypertensive, antifungal [82], and examples of these drugs are included in Table 4.

As it has been mentioned before, the target genes include phase I and II drug-metabolizing enzymes, as well as uptake and efflux drug transporters (Table 4). The expression of these genes can be modified when a PXR ligand binds to the receptor, but an impact of *NR1I2* genotype in the enzyme and transporter induction and/or in the DNA binding has also been observed [80]. The effect of several *NR1I2* variants on different drugs’ metabolism can be found in the literature. For instance: rs3814055 in erythromycin metabolism [83]; rs1464603 and rs1464602 in midazolam clearance [84]; and, rs3814058 and rs2276707 in doxorubicin clearance [85].

**Table 4 viruses-13-00413-t004:** Ligands and target genes of the nuclear receptor PXR [80,82,86].

Nuclear Receptor	Drug Ligands	Target Genes
PXR	Amoxicillin, ampicillin, penicillin, cefuroxime, cephalexin, cefradine, sulfamethazine, erythromycin, rifampin, tetracycline, topiramate, carbamazepine, phenytoin, valproic acid, terbinafine, griseofulvin, clotrimazole, miconazole, nifedipine, cyclophosphamide, cisplatin, docetaxel, paclitaxel, vinblastine, troglitazone, rosiglitazone, atorvastatin, simvastatin, efavirenz, nevirapine, ritonavir, omeprazole, lansoprazole	*CYP2B6, CYP2C8, CYP2C9, CYP2C19, CYP3A4, CES2, UGT1A1, UGT1A6, ABCB1, ABCC2*

CAR is encoded by the *NR1I3* located in chromosome 1, and it consists of nine exons. The exons 2, 3, and 4 determine the DNA binding domain, while the ligand-binding domain is encoded by the sequence of DNA comprised between the end of the exon 4 and the beginning of the 9 [78]. CAR forms a heterodimer with retinoid X receptor that binds to retinoic acid response elements and activates target genes. It shares with PXR significant cross-talk in both target gene recognition by binding to the similar xenobiotic responsive elements in their target gene promoters and accommodating a diverse array of xenobiotic activators. CAR target genes include *CYP2B6, CYP2C8, CYP2C9, CYP2C19, CYP3A4, UGT1A1, ABCB1, ABCC2, ABCC3,* and *ABCC4* [86]. Although this receptor has been less studied than PXR, several genetic variants affecting the DNA and ligand-binding domains have been described [78].

### 6.3. CYP450 Enzymes’ Expression in Inflammation and Infection Processes

Although inflammation and/or infection have not been commonly considered in pharmacogenetic studies, both processes are associated with decreased hepatic expression and/or activities of hepatic and extrahepatic CYP enzymes, drug metabolism, and drug transporters, resulting in a disturbance of the bioavailability of oral drugs [87,88]. Furthermore, the regulation of CYP enzymes’ expression mediated by several cytokines has been reported. In addition, mRNAs down-regulation of several CYPs by interleukin-6 have been observed in human hepatocytes [89], and this interleukin plays a crucial role in the cytokine storm of COVID-19 [90]. Moreover, *IL6* and *IL6R* variants have been recently proposed as prognostic and pharmacogenetic biomarkers of COVID-19, mainly for monoclonal antibodies targeting IL6 and IL6R [91].

Standard dosages in patients with the infectious and inflammatory process, as COVID-19, could increase exposure to the drugs, resulting in a higher possibility of ADR incidence. Simultaneously, for pro-drugs activated by metabolism, the impairment of P450 activities due to inflammation could reduce their therapeutic efficacy. In this sense, inflammatory markers and genes related to the immune response could also be considered in evaluating the inter-individual variability in the responses to COVID-19 treatment.

## 7. Discussion

Several pharmacogenetic biomarkers related to the metabolic pathway of drugs used for COVID-19 treatment have been described in the present review. In agreement with previous reports [7,8,9], there are variants in *CYP2C8*, *CYP2D6*, *CYP3A4*, *CYP3A5*, *SLCO2B1*, *ABCB1*, *ABCC2*, *CES1*, and *G6PD* that could help to improve the clinical outcome of the COVID-19. The scientific evidence supports the study of variants in *CYP2D6, CYP3A4*, *SLCO2B1*, *ABCB1,* and *ABCC2* with the response to specific drugs (Figure 2). Nevertheless, the remaining pharmacogenes should not be discarded because the recommendations and association results are substrate-depending [92], and there is an important influence of the ethnic origin of the studied population [93].

In addition, it is necessary to consider that the drug response results from the gene-environment interaction, in which nongenetic factors (e.g., age, gender, co-treatment, disease severity) must be considered in the pharmacogenetic studies [94]. In this sense, drug-drug interactions and the inflammation and infection processes in COVID-19 could represent relevant sources of therapeutic failures and/or drug toxicity; thus the pharmacogenetic studies should identify the impact of these factors in drug response to determine the precise influence of genetic variants in the COVID-19 treatment [95,96].

## 8. Conclusions

Pharmacogenetics provides insight for the treatment improvement of several diseases, particularly for those treated with drugs presenting a wide inter-individual variability. Several pharmacogenetic markers could be evaluated in the COVID-19 treatment, which is currently based on antivirals, antibiotics, antiparasitics, and/or anti-inflammatory drugs previously used for other infectious and non-infectious diseases. Nevertheless, there are characteristics of the complex disease and the pharmacogenetic biomarkers that should be considered in the design of pharmacogenetic studies of COVID-19. Prospective studies, preferably, besides adequate control of the disease and treatment variables, could lead to valid results for treatment recommendations on the way to personalized therapy in COVID-19.

Besides, future pharmacogenetic markers should be identified for the drugs designed explicitly for the SARS-CoV-2, in which the evaluation of the virus variants in the drug response is warranted.

## Figures and Tables

**Figure 1 viruses-13-00413-f001:**
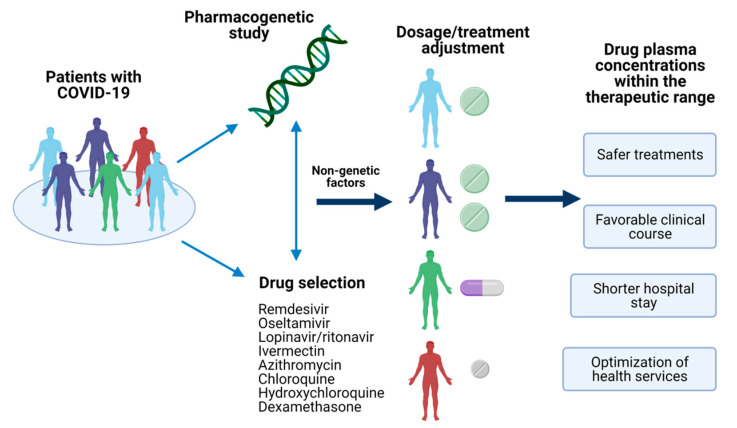
Pharmacogenetics of COVID-19 treatment. The identification of pharmacogenetic biomarkers of relevance in drugs used for COVID-19 treatment and nongenetic factors could provide the information to make dosages adjustments or the selection of the optimal treatment for the patient. Achieving a personalized therapy would assure drug plasma concentrations within the therapeutic range, leading to several advantages in the disease’s clinical outcome. Created with BioRender.com, accessed on 19 February 2021.

**Figure 2 viruses-13-00413-f002:**
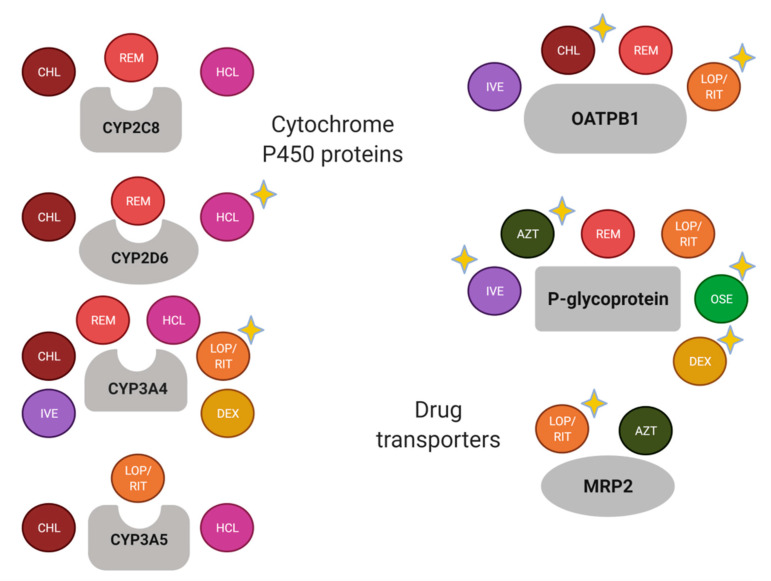
Schematic summary of drug-metabolizing enzymes and transporters of drugs used for the COVID-19 treatment. A yellow-star has been added in drugs with relevant pharmacogenetic knowledge considering if the enzyme or transporter meets the following: (1) it is considered as a Pharmacogenomic Biomarker according to the FDA (https://www.fda.gov/drugs/science-and-research-drugs/table-pharmacogenomic-biomarkers-drug-labeling, accessed on 19 February 2021); (2) it is included as pharmacogene variant in Pharmgkb (https://www.pharmgkb.org/, accessed on 19 February 2021); and/or (3) it has been associated with the pharmacokinetics and/or pharmacodynamics of the corresponding drug in a scientific report. AZT, azithromycin; CHL, chloroquine; DEX, dexamethasone; HCL, hydroxychloroquine; IVE, ivermectin; LOP, lopinavir; OSE, oseltamivir; REM, remdesivir; RIT, ritonavir. Created with BioRender.com, accessed on 19 February 2021.

**Table 1 viruses-13-00413-t001:** Relevant pharmacogenetic variants in cytochrome P450 enzymes ^1^.

Pharmacogene	Variants	Clinical Function
*CYP2C8*	**1A*	Normal function
	**5*, **7*, **11*	No function
*CYP2D6*	**1*, **2*, **27*, **33*, **34*, **35*, **39*, **45*, **46*, **48*, **53*	Normal function
	**9*, **10*, **17*, **29*, **41*, **49*, **50*, **54*, **55*, **59*	Decreased function
	**3*, **5*, **6*, **7*, **8*, **11*, **12*, **13*, **14*, **15*, **18*, **19*, **20*, **21*, **31*, **36*, **38*, **40*, **42*, **44*, **47*, **51*, **56*, **57*, **60*, **62, *68*, **69*, **81*, **92*, **96*, **99*, **100*, **101*, **114*, **120*, **124*, **129*	No function
*CYP3A4*	**1A*	Normal function
	**8*, **11*, **13*, **16A*, **17*, **22*	Decreased function
*CYP3A5*	**1A*, **1D*	Normal function
	**3*, **6*, **7*	No function

^1^ Data from Pharmacogene Variation Consortium at pharmvar.org, accessed on 19 February 2021.

**Table 2 viruses-13-00413-t002:** Pharmacogenetic variants described for dexamethasone response ^1^.

Pharmacogene	Name	Variant	Drug Phenotype
*ABCB1*	ATP Binding Cassette Subfamily B Member 1	rs2032582 rs1045642	Efficacy ^2^ Efficacy ^2^
*DOK5*	Docking Protein 5	rs117532069	Toxicity ^3^
*SERPINE1*	Serpin Family E Member 1	rs6092	Toxicity ^3^
*LINC00251*	Long Intergenic Non-Protein Coding RNA 251	rs141059755	Toxicity ^3^
*BMP7*	Bone Morphogenetic Protein 7	rs79085477	Toxicity ^3^
*PYGL*	Glycogen Phosphorylase L	rs7142143	Efficacy ^4^
*DOK5*	Docking Protein 5	rs117532069	Toxicity ^3^
*CTNNB1*	Catenin Beta 3	rs4135385	Efficacy ^4^

^1^ Data from PharmGKB (pharmgkb.org, accessed on 19 February 2021), ^2^ Survival in multiple myeloma, ^3^ Risk of osteonecrosis in children with Precursor Cell Lymphoblastic Leukemia-Lymphoma, ^4^ Risk of relapse in children with Precursor Cell Lymphoblastic Leukemia-Lymphoma.

**Table 3 viruses-13-00413-t003:** Examples of drugs reported as substrates, inhibitors and/or inducers of CYP3A4, P-glycoprotein, and OATPB1 ^1^.

Protein	Substrates	Strong Inhibitors	Strong Inducers
CYP3A4	Alfentanil, avanafil, buspirone, conivaptan, darifenacin, darunavir, ebastine, everolimus, ibrutinib, lomitapide, lovastatin, midazolam, naloxegol, nisoldipine, saquinavir, simvastatin, sirolimus, tacrolimus, tipranavir, triazolam, vardenafil, budesonide, dasatinib, dronedarone, eletriptan, eplerenone, felodipine, indinavir, lurasidone, maraviroc, quetiapine, sildenafil, ticagrelor, tolvaptan	Boceprevir, cobicistat, danoprevir and ritonavir, elvitegravir and ritonavir, grapefruit juice, indinavir and ritonavir, itraconazole, ketoconazole, lopinavir and ritonavir, paritaprevir and ritonavir, posaconazole, ritonavir, saquinavir and ritonavir, telaprevir, tipranavir and ritonavir, telithromycin, troleandomycin, voriconazole	Apalutamide, carbamazepine, enzalutamide, mitotane, phenytoin, rifampin, St. John’s wort
P-gp	Dabigatran etexilate, digoxin, fexofenadine	Amiodarone, carvedilol, clarithromycin, dronedarone, itraconazole, lapatinib, lopinavir and ritonavir, propafenone, quinidine, ranolazine, ritonavir, saquinavir and ritonavir, telaprevir, tipranavir and ritonavir, verapamil	-
OATPB1	Asunaprevir, atorvastatin, bosentan, danoprevir, docetaxel, fexofenadine, glyburide, nateglinide, paclitaxel, pitavastatin, pravastatin, repaglinide, rosuvastatin, simvastatin acid	Atazanavir and ritonavir, clarithromycin, cyclosporine, erythromycin, gemfibrozil, lopinavir and ritonavir, rifampin, simeprevir	-

^1^ Data from Drug Development and Drug Interactions: Table of Substrates, Inhibitors and Inducers, US Food and Drug Administration. Available in https://www.fda.gov/drugs/drug-interactions-labeling/drug-development-and-drug-interactions-table-substrates-inhibitors-and-inducers#table3-1, accessed on 19 February 2021.

## Data Availability

Not applicable.

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
