# Peer review of "Pharmacogenetics Approach for the Improvement of COVID-19 Treatment"

_viruses, 2021, doi:10.3390/v13030413_

Round 1

Reviewer 1 Report

The short review presents a comprehensive report on the search in pharmacogenomic databases and pharmacogenomic guidelines on the relation between genetic variants and the drugs currently prescribed for the COVID-19 treatment. The study is well organise and presented, a suggestion for the authors is to introduce the relation between genomic variants of SARS-CoV-2 and the possible effects (enhancing/reducing the effect) on the drugs used for the treatments.

Author Response

We thank the Reviewer for the comment. The targets of the drugs included in the present study are not precisely the virus SARS-CoV-2. Nevertheless, it is an interesting concern to consider for antivirals developed specifically for SARS-CoV-2 inhibition. A paragraph mentioning the relevance of genetic variants of the SARS-CoV-2 genome in drugs' therapeutic effect has been added as a perspective in the Conclusion section (Page 11).

Reviewer 2 Report

The manuscript titled “Pharmacogenetics approach for the improvement of COVID-19 treatment” is a review on a contemporary topic.

There are some major comments that need to be addressed:

  1. The goals of the manuscript in abstract (page 1, lines 13-16) and introduction (page 2, lines 57-64) are vague and non-specific.
  2. In this pharmacogenetic manuscript authors didn’t utilize the key terminologies and concepts such as poor metabolizers, extensive metabolizers etc. This is a major drawback.
  3. Page 2, lines 76-77, the sentence is contradictory: “extensively metabolized” vs “predominantly”
  4. Page 4, section 2.3., is oseltamivir a drug used in COVID-19? Please cite the appropriate reference or remove this.
  5. Subsection 3: Do P-gp and MRP2 have any substantial role in azithromycin pharmacokinetics?
  6. Figure 1: The contribution of enzymes for each drug should be demonstrated here. In several cases, the indicated enzyme is not known to have any major contribution. For example, there are no conclusive reports of CYP2D6 involvement in remdesivir metabolism.
  7. Table 3: What’s the need or purpose of this general table in this manuscript?
  8. Page 9, Section 6.3: The discussion on inflammation and infection is totally disconnected with the general theme of the manuscript
  9. A figure highlighting the impact of pharmacogenetics on the pharmacokinetics of the COVID-19 drugs is needed.
  10. The manuscript suffers from the general discussion without any critical comment on the effect of pharmacogenetics on pharmacokinetics and subsequently safety/toxicity of the COVID-19 drugs. Authors need to make significant updates in this direction.

Author Response

1. The goals of the manuscript in abstract (page 1, lines 13-16) and introduction (page 2, lines 57-64) are vague and non-specific.

Authors response: The aim of the review has been modified in both Sections in order to present it in a more specific and consistent manner (Pages 1 and 2).

2. In this pharmacogenetic manuscript, authors didn't utilize the key terminologies and concepts such as poor metabolizers, extensive metabolizers etc. This is a major drawback.

Authors response: Thank you for your observation; the possible extrapolation of CYP genotype to metabolic phenotypes depends on the specific cytochrome in each study. In this sense, for the current manuscript, the terms of poor, extensive, and ultrarapid metabolizers were stated for CYP2D6 (Page 3, 3rd paragraph).

3. Page 2, lines 76-77, the sentence is contradictory: "extensively metabolized" vs "predominantly"

Authors response: That's right, to avoid confusion, the sentence has been modified to: "CYP2C8, CYP2D6, and CYP3A4  are involved in the metabolism of remdesivir, but it is considered that it is predominantly metabolized by hydrolases" (Page 3, 2nd paragraph).

4. Page 4, section 2.3., is oseltamivir a drug used in COVID-19? Please cite the appropriate reference or remove this.

Authors response: Thank you for highlight this interesting point. Our current Review manuscript did not aim to present recommendations for the treatment of COVID-19. The pharmacogenetic variants are described for those drugs that have been used since the pandemic started. Oseltamivir was used in some patients with COVID-19 (references are listed below), particularly at the pandemic's onset, although its effectiveness has been controversial. This has been stated now on page 5, and the corresponding references have been included.

  • Tan Q, Duan L, Ma Y, et al. Is oseltamivir suit-able for fighting against COVID-19: in silico assessment, in vitro and retrospective study. BioorgChem.2020;104:104257.
  • Chiba, S. Effect of early oseltamivir on outpatients without hypoxia with suspected COVID-19. Wien Klin Wochenschr (2020). https://doi.org/10.1007/s00508-020-01780-0
  • Wang D, Hu B, Hu C, Zhu F, Liu X, Zhang J, Wang B, Xiang H, Cheng Z, Xiong Y, Zhao Y, Li Y, Wang X, Peng Z. Clinical Characteristics of 138 Hospitalized Patients With 2019 Novel Coronavirus-Infected Pneumonia in Wuhan, China. JAMA. 2020 Mar 17;323(11):1061-1069. doi: 10.1001/jama.2020.1585. PMID: 32031570; PMCID: PMC7042881.
  • IMU-838 and Oseltamivir in the Treatment of COVID-19 (IONIC). Available in https://clinicaltrials.gov/ct2/show/NCT04516915 (Accessed on February 21, 2021).

5. Subsection 3: Do P-gp and MRP2 have any substantial role in azithromycin pharmacokinetics?

Authors response: Previous studies have reported an implication of ABCB1 variants in azithromycin pharmacokinetics (He et al., 2009; Nazir et al., 2020); actually, it has been considered that the transporter plays an essential role in the synergistic effect of the hydroxychloroquine-azithromycin combination in COVID-19 therapy (Scherrmann, 2020). Meanwhile, MRP2 also has been considered to be involved in the excretion of azithromycin, but to a lesser extent in comparison to P-gp (Sugie et al., 2004).

  • He XJ, Zhao LM, Qiu F, Sun YX, Li-Ling J. Influence of ABCB1 gene polymorphisms on the pharmacokinetics of azithromycin among healthy Chinese Han ethnic subjects. Pharmacol Rep. 2009 Sep-Oct;61(5):843-50. doi: 10.1016/s1734-1140(09)70140-9.
  • Nazir S, Adnan K, Gul R, Ali G, Saleha S, Khan A. The effect of gender and ABCB1 gene polymorphism on the pharmacokinetics of azithromycin in healthy male and female Pakistani subjects. Can J Physiol Pharmacol. 2020 Aug;98(8):506-510. doi: 10.1139/cjpp-2019-0569.
  • Scherrmann JM. Intracellular ABCB1 as a Possible Mechanism to Explain the Synergistic Effect of Hydroxychloroquine-Azithromycin Combination in COVID-19 Therapy. AAPS J. 2020 Jun 12;22(4):86. doi: 10.1208/s12248-020-00465-w.
  • Sugie M, Asakura E, Zhao YL, Torita S, Nadai M, Baba K, Kitaichi K, Takagi K, Takagi K, Hasegawa T. Possible involvement of the drug transporters P glycoprotein and multidrug resistance-associated protein Mrp2 in disposition of azithromycin. Antimicrob Agents Chemother. 2004 Mar;48(3):809-14.

6. Figure 1: The contribution of enzymes for each drug should be demonstrated here. In several cases, the indicated enzyme is not known to have any major contribution. For example, there are no conclusive reports of CYP2D6 involvement in remdesivir metabolism.

Authors response: Thank you for your comment. Figure 1 (now figure 2) has been modified. Considering the pharmacogenetics approach of the present manuscript and the Reviewer's request, a mark has been added in those drugs for which there is pharmacogenetic evidence that the corresponding enzyme or transporter impacts the pharmacokinetics and/or pharmacodynamics of the drugs. The pharmacogenetic evidence was considered if at least one of the following criteria was accomplished: 1) It is considered as a Pharmacogenomic Biomarker according to the FDA (https://www.fda.gov/drugs/science-and-research-drugs/table-pharmacogenomic-biomarkers-drug-labeling); 2) There are included as pharmacogene variant in Pharmgkb (https://www.pharmgkb.org/); and/or 3) If there is a scientific study reporting the association of pharmacogenetic variants with the pharmacokinetics and/or pharmacodynamics of the corresponding drug.

7. Table 3: What's the need or purpose of this general table in this manuscript?

Authors response: COVID-19 is a complex disease that has been treated with a combination of different drugs, and several co-morbidities have been identified as risk factors for a severe form of the disease. According to the present Review information, several drugs used for COVID-19 treatment are metabolized/transported by enzymes and transporters, which presents several drug-drug interactions such as CYP3A4 and ABCB1. Therefore, these interactions are crucial non-genetic factors to consider in the design or pharmacogenetic studies of COVID-19 treatment. In this sense, Table 3 provides handy information about drugs that must be included as a co-variable in the association studies of pharmacogenetic biomarkers with the variability of the pharmacokinetic parameters of the drugs used for the COVID-19 treatment. Now, this has been clarified on Page 8.

8. Page 9, Section 6.3: The discussion on inflammation and infection is totally disconnected with the general theme of the manuscript

Author Response: We apologize for this issue. The impairment of drug-metabolizing enzymes and transporters observed in inflammation and infection processes could cause inter-individual variability to drug response. COVID-19 is an infectious disease in which the inflammation process plays a crucial role in the disease severity. Therefore, considering the inflammation level in pharmacogenetic studies could exclude a bias and explain the clinical outcome differences in patients treated with the same drug. Also, taking into account that cytokines regulate CYPs enzymes' expression, it is plausible to think that variants in genes encoding these immunologic proteins could be implicated in response to anti-inflammatory drugs such as dexamethasone. Clarifications have been added in this section (Page 9).

9. A figure highlighting the impact of pharmacogenetics on the pharmacokinetics of the COVID-19 drugs is needed.

Author Response: Figure 1 has been added.

10. The manuscript suffers from the general discussion without any critical comment on the effect of pharmacogenetics on pharmacokinetics and subsequently safety/toxicity of the COVID-19 drugs. Authors need to make significant updates in this direction.

Author Response: A general discussion has been added on page 10.

Round 2

Reviewer 2 Report

Comments on the first version have been addressed adequately. Thanks